# Dual origin of relapses in retinoic-acid resistant acute promyelocytic leukemia

Jacqueline Lehmann-Che[1,2,3,4], Cécile Bally[3,5], Eric Letouzé [3,6,7,8], Caroline Berthier[1,2,3,9], Hao Yuan[1,2,3], Florence Jollivet[1,2,3], Lionel Ades[3,5], Bruno Cassinat[10], Pierre Hirsch[11,12], Arnaud Pigneux[13], Marie-Joelle Mozziconacci[14], Scott Kogan [15], Pierre Fenaux[3,5] & Hugues de Thé [1,2,3,4,9]

Retinoic acid (RA) and arsenic target the t(15;17)(q24;q21) PML/RARA driver of acute promyelocytic leukemia (APL), their combination now curing over 95% patients. We report exome sequencing of 64 matched samples collected from patients at initial diagnosis, during remission, and following relapse after historical combined RA-chemotherapy treatments. A first subgroup presents a high incidence of additional oncogenic mutations disrupting key epigenetic or transcriptional regulators (primarily WT1) or activating MAPK signaling at diagnosis. Relapses retain these cooperating oncogenes and exhibit additional oncogenic alterations and/or mutations impeding therapy response (RARA, NT5C2). The second group primarily exhibits FLT3 activation at diagnosis, which is lost upon relapse together with most other passenger mutations, implying that these relapses derive from ancestral pre-leukemic PML/RARA-expressing cells that survived RA/chemotherapy. Accordingly, clonogenic activity of *PML/RARA*-immortalized progenitors ex vivo is only transiently affected by RA, but selectively abrogated by arsenic. Our studies stress the role of cooperating oncogenes in direct relapses and suggest that targeting pre-leukemic cells by arsenic contributes to its clinical efficacy.

[1] INSERM U944, Equipe Labellisée par la Ligue Nationale contre le Cancer, Institut Universitaire d'Hématologie (IUH), 75010 Paris, France. [2] CNRS UMR 7212, Institut Universitaire d'Hématologie (IUH), 75010 Paris, France. [3] Univ Paris Diderot, Sorbonne Paris Cité, Institut Universitaire d'Hématologie (IUH), 75010 Paris, France. [4] AP-HP, Unité d'Oncologie Moléculaire, Hôpital St Louis, 75010 Paris, France. [5] AP-HP, Service d'Hématologie Senior Hôpital St. Louis, 75010 Paris, France. [6] INSERM, UMR-1162 Génomique Fonctionnelle des Tumeurs Solides, Equipe Labellisée Ligue Contre le Cancer, Institut Universitaire d'Hématologie (IUH), 75010 Paris, France. [7] Univ Paris Descartes, Labex Immuno-Oncology, Sorbonne Paris Cité, Faculté de Médecine, 75006 Paris, France. [8] Univ Paris 13, Sorbonne Paris Cité, UFR Santé, Médecine, Biologie Humaine, 93000 Bobigny, France. [9] Collège de France, PSL Research University, 75005 Paris, France. [10] INSERM U1131, Institut Universitaire d'Hématologie (IUH) and APHP, Laboratoire de Biologie Cellulaire, Hopital Saint-Louis, Paris 75010, France. [11] Sorbonne Université, Inserm Centre de Recherche Saint-Antoine CRSA, APHP, Hôpital Saint Antoine, 75012 Paris, France. [12] AP-HP, Service d'Hématologie biologique, Hôpital Saint Antoine, 75012 Paris, France. [13] Centre Hospitalier Universitaire (CHU) Bordeaux, Service d'Hématologie Clinique, 33000 Bordeaux, France. [14] Institut Paoli-Calmettes (IPC), Service d'Hématologie biologique, Marseille 13009, France. [15] Department of Laboratory Medicine, University of California, San Francisco (UCSF), San Francisco, CA, USA. These authors contributed equally: Jacqueline Lehmann-Che, Cécile Bally, Eric Letouzé. Correspondence and requests for materials should be addressed to H. de Thé. (email: hugues.dethe@inserm.fr)

Most acute promyelocytic leukemias (APLs) are driven by the t(15,17) translocation that yields the PML/RARA fusion. PML/RARA deregulates transcription, blocking myeloid differentiation and enhancing progenitor self-renewal[1,2]. PML/RARA also disrupts PML nuclear bodies, blunting p53 signaling, impeding senescence, and promoting clonogenic growth[3,4]. Epidemiological studies have supported the view that APL development requires a single rate-limiting step[5]. Yet, in *PML/RARA* transgenic mouse models, leukemia development requires secondary cooperating changes[6–8]. *WT1*, *KRAS*, *NRAS* mutations, *FLT3* activation, or *Myc* trisomy, which are common genetic events in many other subsets of acute myeloid leukemia (AML), may be observed in APL patients[9–14]. These progression events, which occur late in APL or AML development, sharply accelerate PML/RARA-driven transformation in murine models[15–17].

APL is a model for targeted leukemia cure, as two non-chemotherapeutic agents, retinoic acid (RA) and arsenic trioxide (hereafter referred as arsenic), have extraordinary clinical potency and cooperate to eradicate the disease without the need for DNA-damaging chemotherapy[1,18–22]. Retinoic acid and arsenic initiate the degradation of PML/RARA by directly binding to respectively its RARA and PML moieties[18,23]. Importantly, arsenic also targets normal PML—the effector of APL cure[24–26]—likely explaining its extremely potent anti-leukemic effects as a single agent[1,27]. In historical patients whose frontline treatment did not include arsenic, relapse rates were up to 30% (ref. [28]). Some situations of RA resistance may be caused by mutations in the RARA moiety of PML/RARA[29], but the natural history of APL development and resistance to the RA/chemotherapy regimen remains imperfectly understood. Here we show that relapses are associated with the presence of potent PML/RARA cooperating oncogenes at diagnosis, or re-emergence of an ancestral pre-leukemic clone that survived targeted therapy with RA.

## Results

**Exome sequencing of diagnosis and relapse APLs pairs**. To define the pre-existing or acquired mutations associated with RA/chemotherapy resistance, we performed whole-exome sequencing of diagnosis and relapse pairs from 23 patients recruited through the French Swiss Belgian APL group (GTLAP) trials. Complete remission samples were available for 18 patients allowing identification of somatic variants at diagnosis and relapse; the 5 others diagnosis and relapse pairs were used to identify mutations acquired at relapse (patients' features in Supplementary Table 1). We obtained a mean depth of 91×, with on average 88% of target regions covered ≥25×. At diagnosis, we identified 194 non-synonymous substitutions and 32 small insertions/deletions (indels), corresponding to a median of 12.5 protein-coding

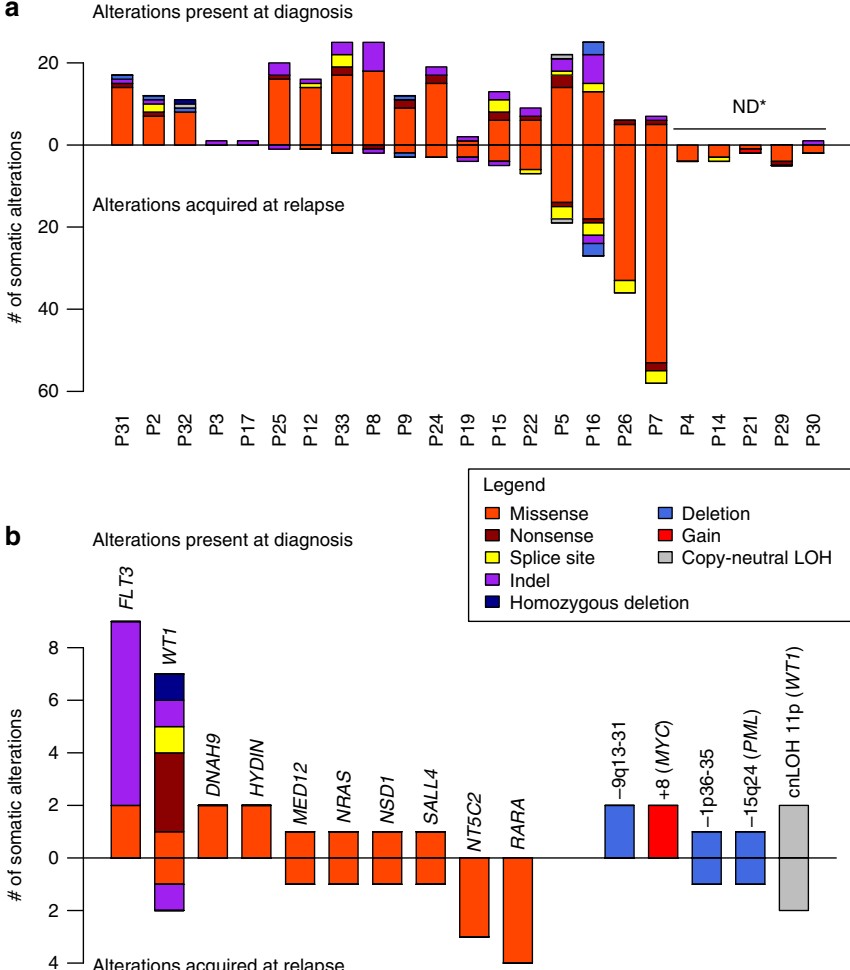

**Fig. 1** Graphic summary of the exome analysis of relapsing APLs. **a** Number and type of somatic alterations identified at diagnosis (upper part) and acquired at relapse (lower part) for each patient. ND* indicates sample pairs with no available remission germline DNA, precluding determination of diagnostic alterations. **b** Somatic mutations (left) and copy-number alterations (right) observed at diagnosis (upper part) or relapse (lower part) at least twice in the study. Note the unexpected high prevalence and molecular variety of *WT1* alterations

mutations per sample, very similar to unselected de novo APLs[11,12] or AMLs[30] (Fig. 1a, complete list of alterations in Supplementary Data 1, presumed drivers in Supplementary Data 2, comparisons with previous studies in Supplementary Table 2). Most of these changes are non-synonymous mutations in genes never implicated in cancer, likely representing passenger mutations acquired before oncogenic activation or early during expansion of PML/RARA clones[12]. At relapse, we only observed a median of three additional genetic lesions, very unevenly distributed among patients (range 0–61, Fig. 1a). These data are in line with previous studies suggestive for a reliable estimation of the mutation burden in APL.

**WT1 is often altered at diagnosis in relapsing APLs.** In non-relapsing APLs, alterations commonly associated to PML/RARA primarily affect *FLT3* (40%), *WT1* (10%), *NRAS* (10%), or *KRAS* (5%)[11,12]. In our relapsing population, these were observed at the expected frequencies (see Fig. 1 and Supplementary Tables 3 and 4 for a summary of recurrent alterations at diagnosis and/or relapse), except for *WT1* mutation or loss (7/18, 40%), significantly more frequently observed at diagnosis than in patients not experiencing relapse ($p < 0.01$ Fisher exact test)[11,31,32] (Fig. 1b and 2). Furthermore, we observed copy-neutral LOH of 11p leading to the duplication of the mutant *WT1* allele in four samples, two present at diagnosis and two acquired at relapse, further stressing importance of *WT1* alterations in favoring therapy resistance (Fig. 1b and Supplementary Table 5).

In addition, we found an overall high incidence of mutations activating the MAPK pathway (*BRAF, KIT, PDGFRA*), or inactivating transcriptional or epigenetic regulators (*NSD1, ASXL1, MED12, KDM6A*), which appear uncommon in APLs that did not undergo relapses[11,12]. We identified copy-number alterations (CNAs), notably gains of chromosome 8 (*MYC* locus)[10,15] (Fig. 1b, Supplementary Fig. 1, 2 and Supplementary Table 6). We also detected loss of normal PML in patient 5 (relapse) and 16 (diagnosis), as well as duplication of the mutant *PML/RARA* allele (patient 9, relapse). Finally, we discovered recurrent mutations (at diagnosis and/or relapse) in oncogenes (*SALL4, MED12, NSD1*) previously linked to RA-signaling (Fig. 1b). Intriguingly, recurrent mutations were also found in *HYDIN, DNAH9*, two genes whose products associate to the primary cilium[12,33,34].

We estimated the cancer cell fraction (CCF), i.e. the proportion of tumor cells harboring each somatic mutation at diagnosis and relapse (Supplementary Figure 3 and Table 2). Only 31 mutations identified in diagnosis samples were considered subclonal, including 4 *FLT3* mutations, 3 of which were lost at relapse (patients P2, P16, and P22) and one became clonal at relapse (P31). Thus, in line with other AML studies, *FLT3* mutations tend to occur late in sub-clones of diagnosis samples and may or may not be present at relapse.

**Recurrent alterations acquired at relapse.** As expected, *RARA* mutations were frequently acquired at relapse (four patients), but we also identified recurrent mutations of *NT5C2* specifically acquired at relapse in three patients (Fig. 1b). This gene, implicated in cytarabine or 6-Mercaptopurine responses, was identified as a driver of relapse in childhood acute lymphoblastic leukemias[35]. Among the 18 patients with complete remission, diagnosis, and relapse samples (Fig. 2), 10 displayed acquisition of new driver mutations at relapse (*RARA*, $n = 4$; *WT1*, $n = 3$; *NT5C2*, $n = 2$; *KRAS, NRAS, TFE3, MED12, CDK12, SALL4, NSD1, KMT2C, MYB, TET2*, $n = 1$). Four patients acquired no additional driver alteration at relapse but displayed one or more potent driver already present in the last common ancestor (*FLT3*,

$n = 2$; *ETV6, STK11, HYDIN, MAP2K1, H3F3A, NIN, NSD1, DCTN1, KDM6A*, $n = 1$). The four remaining patients had no identified driver alteration in the relapse sample. These relapses may be driven by alterations undetectable by whole-exome sequencing or insufficiently covered in our data.

**Clonal evolution models define different relapse patterns.** These genetic markers allowed us to reconstruct unambiguous clonal evolution models (Fig. 2). Nine patients presented with a simple linear evolution where all oncogenic or passenger alterations present at diagnosis were similarly found at relapse (Fig. 2a). Similar to chemotherapy-treated AMLs[36,37], four related cases had evidence for subclonal evolution at relapse, with loss of at least one mutation present at diagnosis and acquisition of additional relapse-specific changes (Fig. 2b). Critically, 11 of these 13 patients harbored, at both diagnosis and relapse, potent cooperating oncogenic mutations that likely precluded efficient clearance of the diagnosis clone. In 8 of these 13 cases, relapses were accompanied by acquisition of new oncogenic mutations (*WT1, NRAS, NSD1, MED12, KRAS, ETV6*) or inactivation of genes directly implicated in RA (*RARA*) or cytarabine (*NT5C2*) responses[35].

In contrast, in the five remaining APLs trios, mutations present at diagnosis, notably FLT3 activation in three of these cases, were no longer detected at relapse (Fig. 2c). While patient 19 presented with a TET2 mutation and patient 16 with *MYB* and *NRAS* mutations, the three others did not exhibit any relapse-specific genomic alterations. Diagnosis and relapses only shared rare passenger mutations. Yet, breakpoints, assessed at the mRNA levels, were identical between diagnosis and relapse. The distinct relapse patterns in these patients are thus suggestive for the existence of a pre-leukemic *PML/RARA*-expressing clone that survived RA/chemotherapy and reinitiated APL (see below).

Analysis of the five diagnosis-relapse only pairs (in which somatic mutations at diagnosis could not be assessed) confirmed the low number of additional mutations at relapse observed in the trios (Fig. 1a). One pair had evidence for a linear evolution (*FLT3/MYC* activation at both diagnosis and relapse, with mutation of *NT5C2* at relapse). The three others did not exhibit new oncogenic changes upon relapse.

Collectively, diagnosis *WT1*, epigenetic or kinase mutants were generally retained in the relapse APL clones and favored the emergence of additional driver or resistance mutations. In contrast, in cases where FLT3 was the only cooperating mutation, it was often lost at relapse and the disease reinitiated from a pre-leukemic clone.

**Arsenic, but not retinoic acid target self-renewal in APL.** That some relapses have very distinct mutational profiles from the diagnosis clone implies that they derive from ancestral pre-leukemic PML/RARA-only expressing cells. However, prolonged RA-therapy, by triggering PML/RARA destruction, should have precipitated their loss. Persistence of pre-leukemic AML clones in remissions after chemotherapy was reported[38]. Yet, PML/RARA expressing cells were repeatedly undetectable in remission. We therefore explored ex vivo *RARA*- or *PML/RARA*-transformed mouse progenitors in methylcellulose cultures, examining their clonogenic potential upon RA-exposure and subsequent drug withdrawal. In *RARA*-transformed cells, RA definitively abolished clonogenic growth (Fig. 3a). In contrast, in *PML/RARA*-transformed progenitors, RA only transiently affected growth, as differentiated RA-treated progenitors could reinitiate colony formation upon drug withdrawal[39] (Fig. 3b). Arsenic modestly decreased growth and did not affect self-renewal of *RARA*-

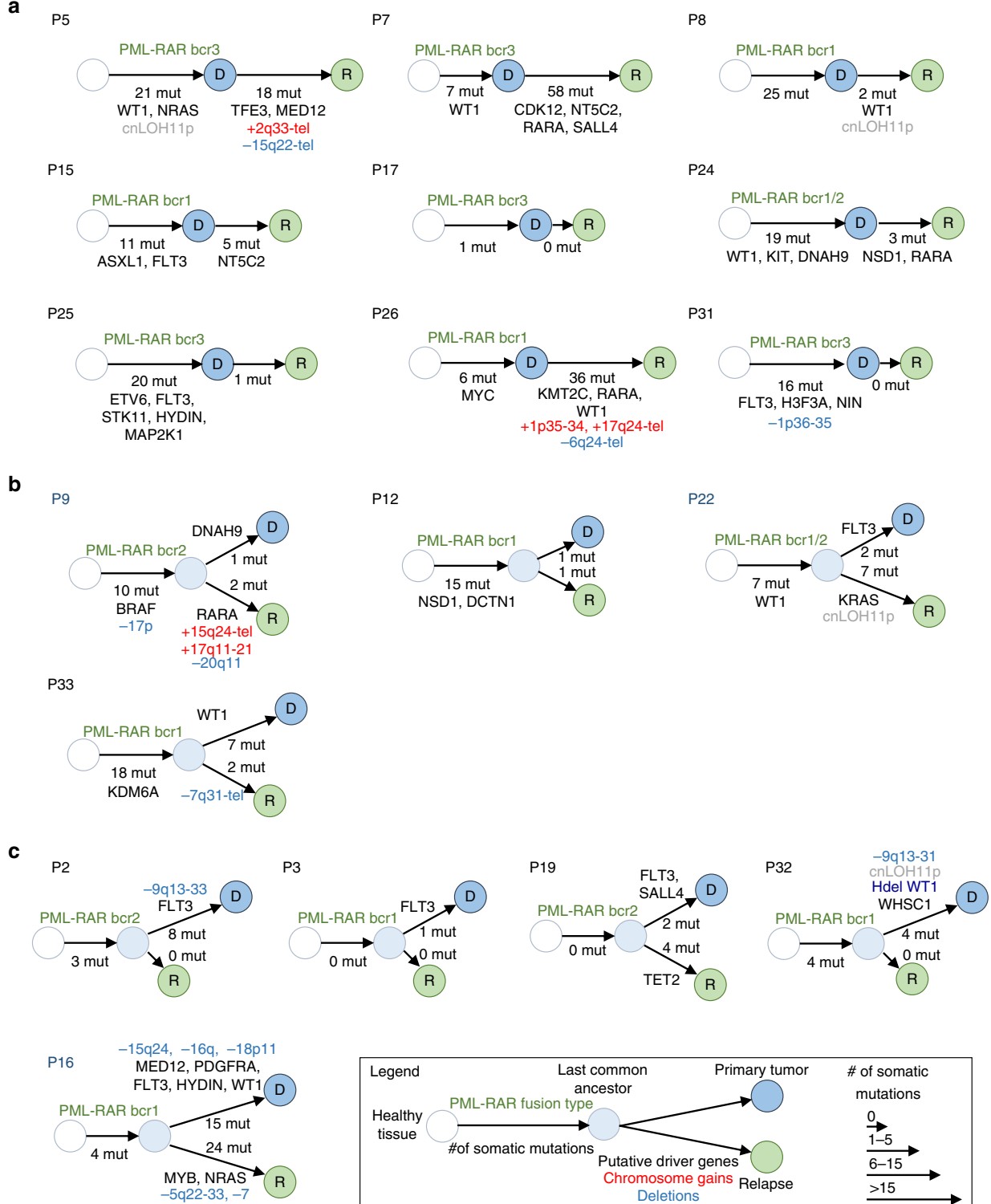

**Fig. 2** Tumor progression trees reconstructed for 18 patients with matched primary tumor and relapse samples. For each patient, a diagram represents the predicted evolution linking the cell of origin (white circle), the PML/RARA-expressing common ancestor (light blue circle), the diagnostic sample (dark blue circle), and the relapse sample (green circle). The number of somatic protein-coding mutations occurring at each step is indicated below the arrows, together with the type of PML/RARA breakpoint, chromosomal gains (red), and deletions (blue), and mutations affecting known driver genes or new recurrent genes identified in this study. The size of arrows is proportional to the number of somatic mutations occurring at diagnosis or relapses. Three major modes of evolution are identified. **a** Nine patients present a linear evolution where all events detected at diagnosis are also present at relapse, with (P5, P7, P8, P15, P24, P25, P26) or without (P17, P31) new acquired alterations. **b** Four patients display branched evolution with many alterations shared by the primary and relapse samples but also specific to one or the other, suggesting that the relapse evolved from a sub-clone of the primary tumor. **c** Five patients displayed no or very few alterations apart from the *PML-RARA* fusion in the relapse samples, suggesting that they emerged from pre-leukemic *PML/RARA*-expressing clones

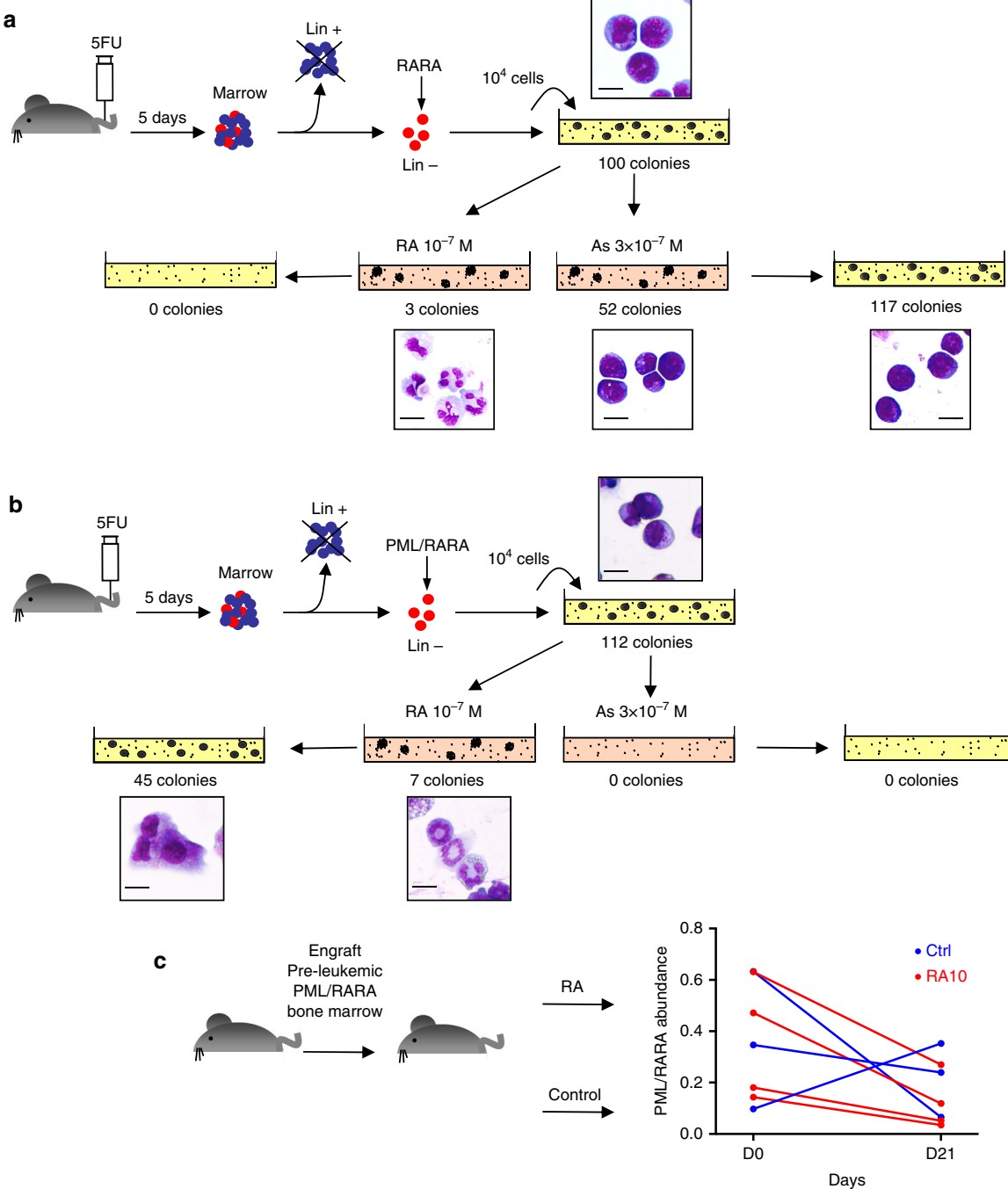

**Fig. 3** Effect of therapies on self-renewal of *RARA*- or *PML/RARA*-transformed progenitors. **a** Mouse primary progenitors were immortalized by *RARA* overexpression and propagated in methylcellulose cultures with the indicated treatments. Treated cells were then regrown without further drug exposure. Colony numbers and MGG stains are shown. Scale bar: 10μm. **b** Same as above with *PML/RARA*-transformed progenitors. Representative experiment of three independent replicates. **c** Effect of RA therapy on the abundance of pre-leukemic *PML/RARA*-expressing cells in vivo. Relative *PML/RARA* DNA abundance, as determined by qPCR comparing *PML/RARA* and *CEBPA* abundance, is indicated for each mouse before and after therapy. Treatment did not result in significant difference in PML/RARA abundance, as demonstrated by paired Student's *t*-test The mouse images in this figure were drawn by Dr. Lehmann-Che for use in this paper

transformed progenitors, but strikingly abolished clonogenic activity of *PML/RARA*-transformed ones (Fig. 3a, b), further demonstrating that arsenic response is mediated through PML[40,41]. We assessed the relevance of these observations in vivo, by engrafting bone marrow from pre-leukemic *PML/RARA* transgenics into irradiated syngenic recipients. When bone marrow chimerism was established, animals were treated for 3 weeks with RA and bone marrow collected. In keeping with the ex vivo

studies, only a small decrease in PML/RARA burden, as detected by quantitative PCR, was observed (Fig. 3c).

## Discussion

Our studies of relapsing patients collected over 20 years bear several important conclusions. First, they highlight the genetic simplicity of APLs, since in these stringently explored patients,

some APLs did not exhibit any PML/RARA cooperating events, although we cannot exclude the existence of non-coding mutations or epigenetic changes. Second, we found that many APLs that will undergo relapse are associated with the presence at diagnosis of a high prevalence of WT1 alteration as well as uncommon mutations affecting activators of MAP kinase pathway and/or other epigenetic regulators (Fig. 2), suggesting that these are responsible for therapy resistance[31]. In that respect, MAP kinase activation blunts p53 response by enhancing HDM2 expression[42], likely opposing the PML/p53-driven senescence program implicated in APL eradication[1,24]. The high incidence of WT1 inactivation, often bi-allelic, stresses the importance of this complex pathway[43] which promotes growth, affects epigenetic regulation[44,45], but also directly influences RA signaling[46]. These and other strong survival/proliferation signals are expected to favor the subsequent selection of PML/RARA mutations associated to RA resistance or other oncogenic mutations, as was indeed observed (Fig. 2). Third, we observed recurrent mutations in key regulators of RA signaling: NSD1, an epigenetic regulator of RA response[47], translocated in rare AMLs and mutated in some solid tumors[48] and in SALL4, a key RA-target in germ cell development[49]. MED12 and RARA (altered in some of our APLs) may also be mutated in phyllode breast cancers[50]. Identification of relapse-specific NT5C2 mutations, previously reported in acute lymphoblastic leukemia[35] but never in APL or non-APL AMLs, genetically demonstrates that cytarabine and/or 6-Mercaptopurine have therapeutic efficacy in APL[28,51]. A previous study explored 8 trios and subsequently investigated 400 loci (genes from their discovery set and others known from the literature) in a large cohort of 200 APLs. Contrasting with the current study, these patients had been treated with very heterogeneous regimen (RA, As and/or chemotherapy) and germline DNA and/or relapse samples were unavailable for many patients. While their conclusions are generally in line with our findings, they did not observed higher WT1 alterations at relapse, but found a high frequency of ARID1A/B or RUNX1 mutations[11] (see Supplementary Table 2 for comparison). This may reflect their high patient heterogeneity and the comparatively smaller patient number from our study.

The most unexpected observation was that some relapses were completely distinct from the diagnostic APL clone. This was previously demonstrated in chemotherapy-treated core-binding factor leukemias[37,52,53]. These APL relapses likely derive from long-lasting pre-leukemic PML/RARA-expressing clones, undetectable in remission bone marrow samples, which resisted prolonged RA therapy[54]. Such retinoic acid-resistance of the clonogenic activity of pre-leukemic cells is directly supported by ex vivo studies (Fig. 3), highlighting the uncoupling between differentiation and loss of self-renewal[2,39]. PML/RARA opposes senescence[55,56], explaining maintained self-renewal upon RA-retrieval. In sharp contrast, arsenic abolishes self-renewal of PML/RARA-, but not RARA-driven pre-leukemic cells, perhaps contributing to its clinical superiority to preclude occurrence of late relapses[21,57–59]. In that respect, most patients received arsenic at relapse and reached complete remissions, except for patient 5, who presented with a deletion in the normal allele of PML at relapse, predicted to impede therapy response[24–26]. Critically, this implies that arsenic therapy can override the survival signals enforced by cooperating oncogenes of type I relapses (Fig. 2a, b). Clinical data (white blood cell count and time to relapse) were not statistically linked to the different types of relapses (Supplementary Table 1), likely reflecting low statistical power within this small population. Collectively, these findings highlight two very distinct modes of APL relapse to historical RA/chemotherapy regimen. They suggest a novel mechanism explaining the clinical activity of arsenic and have broad implications to our understanding of targeted therapies.

## Methods

**Patients**. We identified patients treated in trials of the French Swiss Belgian APL group (APL93 (2 patients), APL2000 (20 patients), and APL2006 (1 patient)) and who had experienced at least one relapse. All patients had received first-line treatment with ATRA + chemotherapy according to the protocols. Patient did not receive any arsenic as induction or consolidation therapy. We retrospectively collected samples from patients at diagnosis, at complete remission and at first relapse except for P22 who was analyzed at second relapse. The ethical review board (CPPRB 2016-04-01) approved the study and informed consent for genomic analyses was obtained from patients, considering absence of opposition after a month as approval. Genomic DNA was retrieved from frozen cells or samples for cytogenetic analyses. DNA was extracted by conventional techniques. Genomic DNA quantity and purity were assessed by Qubit® 2.0 Fluorometer (Invitrogen) and NanoDrop ND-1000 (Thermo Scientific) as well as visual inspection of agarose gel electrophoresis. For whole-exome sequencing, native genomic DNA was fragmented with the Covaris S2 system. Sequencing adaptor ligation was performed using the Agilent SureSelect XT (Agilent Technologies) preparation kit. Subsequently, the libraries were captured using Agilent SureSelect XT Human All Exon v.5 probes (Agilent Technologies) and amplified. After quantification and qualification on a Caliper LabChip GX (Caliper Lifescience), the libraries were sequenced on an Illumina HiSeq 1000 platform (Illumina), 2 × 100 cycles, with TruSeq SBS v3 chemistry. Five trios were discarded on the basis of insufficient quality of one of the samples. The relapse of patient 3, which comprised 10% blasts, did not show evidence for the FLT3 indels in any of the 125 reads of the relapse sample, allowing assignment to the second group of relapses. All alterations of driver genes were controlled by stringent visual inspection of the primary sequencing data. Whenever DNA was left for analysis, several clonal oncogenic alterations were confirmed by Sanger sequencing ($n = 5$) or allelic discrimination ($n = 1$). In all cases, the method was adapted to the sensitivity needed to detect the alteration.

**Sequence alignment and variant calling**. Raw sequence alignment and variant calling were carried out using Illumina CASAVA 1.8 software. CASAVA performs the alignment of reads to the human reference genome (hg19) using the alignment algorithm ELANDv2, and then calls single-nucleotide variants and short insertions and deletions (indels) based on allele calls and read depth. We used an Integragen in-house pipeline to annotate each variant according to its presence in the 1000Genome[60], Exome Variant Server (EVS)[61] or Integragen database, and according to its functional category (synonymous, missense, nonsense, splice variant, frameshift, or in-frame indels). To detect the common FLT3 internal duplications that may be missed by classical variant calling algorithms, we used bam-readcount (https://github.com/genome/bam-readcount) to determine the number of mutated bases in an extended region around the known duplication site (chr13: 28608150-28608349). Samples with mutated bases across the regions were then screened visually using the Integrative Genomics Viewer[62].

**Somatic coding variants at diagnosis and relapse**. We considered only variants located within the exome capture baits and we applied stringent filters to keep only reliable variants sequenced in ≥10 reads, with ≥5 variant calls and a QPHRED score ≥20 for both SNP detection and genotype calling (≥30 for indels). Somatic status was first defined for each leukemic sample (diagnosis and relapse) individually: we considered a mutation to be somatic if the variant allele fraction (VAF) was ≥0.15 in the tumor and <0.05 in the remission sample. To identify variants that may be missed in one of the two tumor samples of a same patient due to clonality or technical differences, we then recovered variants that were detected as somatic in one tumor and displayed a VAF ≥0.05 or at least two mutated reads in the second tumor sample. We excluded known germline variants with a minor allele frequency >1% in 1000Genomes, EVS, or Integragen proprietary database. All mutations in recurrently mutated genes (≥5 overall variants or ≥2 relapse-specific variants) or critical to reconstruct evolutionary trees were validated by stringent visual control using the Integrative Genomics Viewer[62]. We used the Palimpsest R package (https://github.com/FunGeST/Palimpsest) to estimate the CCF of each somatic mutation, i.e., the proportion of tumor cells harboring the mutation, taking into account the VAF and local copy-number estimates, as previously described[63,64]. A mutation was considered subclonal if the upper boundary of the 95% confidence interval of the CCF was smaller than 0.95.

**Copy-number analysis**. To identify CNAs in diagnostic and relapse samples, we identified germline single-nucleotide polymorphisms (SNPs) in each sample and we calculated the coverage log ratio (LRR) and B allele frequency (BAF) at each SNP site. Genomic profiles were divided into homogeneous segments by applying the circular binary segmentation algorithm, as implemented in the Bioconductor package DNAcopy[65], to both LRR and BAF values. We then used the Genome Alteration Print (GAP) method[66] to determine the ploidy of each sample, the level of contamination with normal cells, and the allele-specific copy number of each segment. Chromosome aberrations were defined using empirically determined thresholds as follows: gain, copy number ≥ ploidy +1; loss, copy number ≤ ploidy −1; high-level amplification, copy number > ploidy +2; homozygous deletion, copy number = 0. Finally, we considered a segment to have

undergone LOH when the copy number of the minor allele was equal to 0. All aberrations were validated by visual inspection of the LRR and BAF profiles.

**Reconstructing tumor progression trees.** Tumor progression trees could be reconstructed for 18 patients for which complete remission, diagnosis, and relapse samples were available. We first discarded mutations that were identified in only one of the tumor sample but covered by <10 reads in the other and may thus be undetected for technical reasons. We then classified somatic mutations and CNAs in three categories: events common to the diagnosis and relapse samples (that were thus acquired early in a common ancestor of the two clones), events specific to the diagnosis, and events specific to the relapse that were acquired late after the separation of the two clones. For nine patients, all mutations present at diagnosis were also present at relapse so the last common ancestor was the diagnosis clone itself. Of note, *PML-RARA* fusions were common to diagnosis and relapse in all patients and were thus early events occurring in the last common ancestor.

**Mouse studies.** *RARA-* and *PML/RARA*-immortalization of primary hemato-poietic progenitors were performed as previously[39,67]. Briefly, lineage-depleted mouse bone marrow hematopoietic cells collected from 5-fluorouracil-treated mice (3 mg) were infected with retroviruses obtained by transient transfection of Plat-E cells with pMSCV-*PML-RARA*. After spinoculation, we cultured transduced cells in methylcellulose medium (Stem Cell Technologies, M3231) supplemented with 100 ng/ml stem cell factor and 10 ng/ml each of interleukin IL-3, IL-6 and gran-ulocyte/macrophage colony-stimulating factor (Stem Cell Technologies). After a week, we recovered neomycin-selected cells from methylcellulose and replated them at 10,000 cells per well. Treatment with $10^{-7}$ M RA or $3 \times 10^{-7}$ M arsenic trioxide were performed by mixing the drug with the methylcellulose media. After a week, colonies were counted and cells were regrown in fresh media without treatment. Differentiation was morphologically assessed on MGG-stained cells.

Frozen bone marrows from *MRP8-PML/RARA* transgenics[68] were injected in lethally irradiated FVB-strain male syngenic mice of 8 weeks of age ($n = 7$). After a month and complete hematopoietic restoration, bone marrow was taken by femoral puncture and chimerism assessed by qPCR comparing abundance of the *PML/RARA* transgene and *CEBPA*, a gene common to transgenic and host cells. Mice were then randomized to be treated or not by slow release 10 mg RA tablets (Innovative Research of America) and bone marrow was drawn after 3 weeks for a new determination of *PML/RARA*-positive cells. Animals were handled according to the guidelines of institutional animal care committees, using protocols approved by the "Comité d'Ethique Experimentation Animal Paris-Nord" (no. 121).

**Data availability.** Exome data from the 64 samples used in this study have been deposited at the European Genome-phenome Archive (EGA), which is hosted at the EBI and the CRG, under accession number EGAS00001002893. The authors declare that all the other data supporting the findings of this study are available within the article and its supplementary information files and from the corre-sponding author upon reasonable request.

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

## Acknowledgements

The H.d.T. laboratory is supported by the Ligue Nationale contre le Cancer, INSERM, CNRS, University Paris Diderot, Collège de France, Institut National du Cancer, the Association pour la Recherche contre le Cancer (Prix Griffuel), European Research Council Senior Grant 268729-STEMAPL (to H.d.T.), TRANSCAN (DRAMA project), French National Research Agency "Investissements d'Avenir" Programs ANR-11-PHUC-002 and ANR-10-IHUB-0002. We warmly thank medical doctors who provided samples from patients and clinical data: J.P. Marolleau, M. Uzunov, C. Gervais, N. Gachard, E. Delabesse, E. Lippert, S. Raynaud, H. Lapillonne, M. Hunault, C. Ferrand, A. Guerci, F. Mugneret. We thank A. Bazarbachi for critical reading of the manuscript. We thank the genomic platform of IUH and Integragen for sequencing, GeCo for extensive data analysis, the molecular oncology unit for preparing the samples.

## Author contributions

P.F., H.d.T. conceived the study; J.L.-C., C.Ba., P.H., B.C. collected tumor specimens and supervised sample preparation; J.L.-C., H.d.T. supervised whole-exome sequencing; E.L. performed the bioinformatics analysis; P.F, C.Ba., L.A. supervised clinical protocols and data analysis; C.Be., H.Y., F.J. performed experiments on leukemic progenitors, P.H., A.P., M.J.M. provided patient samples and clinical data. S.K. provided murine bone marrows and analyzed data, H.d.T. wrote the manuscript with the assistance and final approval of all authors.

## Additional information

**Competing interests:** The authors declare no competing interests.

