## [Peer Review File · Nature Communications]

Reviewers' comments:

Reviewer #1 (Remarks to the Author):

This is an interesting manuscript which performs mutational analysis to understand a potential genetic basis for relapse in acute promyelocytic leukemia (APL). Given that there have been few studies of genetic alterations in APL accompanying the PML-RARA fusion, this represents an important effort. However, despite the relatively straightforward experimental approach of this study, the results are presented in a somewhat confusing manner and it is not clear what the genetic basis is for relapse in most cases (at least based on how the data are presented).

-The manuscript does not provide enough clarity on the order of mutation acquisition in APL relative to acquisition of the PML-RARA fusion. Some greater effort to provide an explanation of which mutations occur at each stage of disease with greater clarity than shown in Figure 2 is needed. Figure 2 is confusing as there is a white circle which is shown to have accumulated mutations predating the blue circle where PML-RARA is presumably present. It is not clear how the authors determined the order of mutations relative to each other and relative to acquisition of the PML-RARA fusion. Also, it is not clear if the numbers written on the line between the blue and green circle describe how many mutations were gained and lost between diagnosis and relapse.

-Due to these issues above it is really not clear that the authors have enough data to justify the title "dual origin of relapses." Moreover, the title of the paper is also ambiguous as neither the cell-of-origin or molecular basis for relapse are explained by the data shown in the manuscript.

-For the samples that relapse without FLT3 or other mutations what is the basis for relapse? Is the PML-RARA fusion present? This is not clear from the text or figure and seems to be a major issue in the manuscript.

-The Figure legend for Figure 2 needs to more precisely describe what the "major modes of evolution" are as it is not very clear from the figure or text.

-Where are the remission samples on Figure 2 and do the authors know if the PML-RARA fusion was absent at remission?

-It is not clear why or how WT1 mutations or other genetic alterations found at relapse in APL might functionally promote resistance to ATRA and/or arsenic. It is also not clear if these mutations actually promote drug resistance or are merely enriched in relapse.

-It is very unclear how Figure 3 relates to the rest of the manuscript. Moreover, Figure 3a and 3b are largely schematic and it is necessary to present the data in more terms with information on biological and technical replicates and statistical evaluation of the data.

Reviewer #2 (Remarks to the Author):

A interesting paper in which the genomics of APL are examined by next generation sequencing. There has been extensive work on the genetic heterogeneity of diagnosis and relapse of AML in general but less on APL. The work is well done and some novel genes described particularly at relapse. The series is larger than that of Madan et al but the latter study looked at a panel of nearly 400 genes at diagnosis and relapse in a larger set of patients and as such has more power than this work at least for the panel of genes studied by the Koeffler group. Not enough reference to this prior work is given and the results are not compared and contrasted with Madan et al. The work in the present manuscript on clonal evolution from the deep sequencing is interesting and the finding of many subclones at diagnosis, that are enriched at relapse adds new information. The presence at a new clone at relapse not found at diagnosis suggests the persistence of an ancestral PML-RAR positive clone. This is the heart of the novelty of the work. The ex vivo studies with treatment of PML-RAR transformed murine marrow with ATRA show that clonogenic cells can persist, but note that was shown previously from transfer of ATRA treated cells from one mouse to another by the authors previously.

Page 4 bottom "...distinctly uncommon in unslected APLs." I am not sure if the authors mean that these mutations NSD1, MED12 are uncommon in AML versus APL or are they referring to the Madaan paper which did not describe these mutations in their series of 12 exomes.

In all the work does add some new information, mostly on clonal evolution and perhaps some other genes not before identified in APL or for that matter AML as mutated such a HYDIN.

Reviewer #3 (Remarks to the Author):

The authors perform exome sequencing in 64 matched diagnosis, remission and relapse samples from 23 patients treated with RA/chemotherapy protocols. Two subgroups are identified - one with oncogenic mutations disrupting key epigenetic and transcriptional regulators, while relapse from a second subgroup exhibited FLT3 activation. The relapses associated with this subgroup were lost, together with most other mutations, and the authors posit that these relapses derive from pre-leukemic PML/RARA expression cells.

Overall, this is an interesting study. However, there are a number of major issues that need to be addressed before it can be considered for publication:

1). The authors obtained a mean depth of 91X - however, on average only 88% was covered >25X. The authors should perform power calculations to ascertain the likelihood of missing mutations. Relatedly, it would be useful to explore the extent to which their cohort differs from what is expected. To what extent can the low mutation burden be attributed to the low coverage? This has important implications for the conclusions of the study.

2). In general, it would be useful to put their results directly in context of previously published findings., e.g. <https://www.ncbi.nlm.nih.gov/pmc/articles/PMC4972641/> and/or mutations from related cancers in TCGA.

3) The authors boast that they have 'stringently validated genetic markers'. However, it is unclear from the methods how stringent their validation process was. Specifically, the authors state that 'whenever DNA was left for analysis, oncogenic alterations were successfully confirmed by Sanger sequencing'. Without providing further information, it is very difficult to assess how reliable these alterations are.

4). For phylogenetic analysis, the authors should consider the power they have to detect variants in each sample. For example, if an alteration is present in relapse but not diagnosis, were the authors powered to detect it in the diagnosis sample? I would consider setting a lower threshold than <5% when it has already been identified in the relapse.

5) For Figure 1 - it is unclear which mutations were only found at diagnosis, and which were shared between diagnosis and relapse.

6) The authors should consider analysing their data in greater depth to obtain accurate portraits of the clonal structure of each tumor. For instance, further analysis of VAFs and local copy number (as in e.g. Landau, 2014) would allow the cancer cell fraction of mutations to be assessed. This would greatly improve the analysis.

7) GAP is designed for SNP copy number. The authors may therefore wish to also use an alternative copy number tools, such as Sequenza (Favero, 2015), which is explicitly designed for exome sequencing data. It would be useful to include plots of the copy number for the different tumours and also evaluate the extent to which the copy number is identical.

8) In general, the bioinformatics methods were not terribly clear in the manuscript, and should be made more detailed.

Reviewers' comments:

Reviewer #1 (Remarks to the Author):

This is an interesting manuscript which performs mutational analysis to understand a potential genetic basis for relapse in acute promyelocytic leukemia (APL). Given that there have been few studies of genetic alterations in APL accompanying the PML-RARA fusion, this represents an important effort. However, despite the relatively straightforward experimental approach of this study, the results are presented in a somewhat confusing manner and it is not clear what the genetic basis is for relapse in most cases (at least based on how the data are presented).

-The manuscript does not provide enough clarity on the order of mutation acquisition in APL relative to acquisition of the PML-RARA fusion. Some greater effort to provide an explanation of which mutations occur at each stage of disease with greater clarity than shown in Figure 2 is needed. Figure 2 is confusing as there is a white circle which is shown to have accumulated mutations predating the blue circle where PML-RARA is presumably present. It is not clear how the authors determined the order of mutations relative to each other and relative to acquisition of the PML-RARA fusion. Also, it is not clear if the numbers written on the line between the blue and green circle describe how many mutations were gained and lost between diagnosis and relapse.

We apologize if the clarity of our explanations was insufficient and we have clarified these points in the revised manuscript.

To reconstruct the tumor progression trees represented in Figure 2, we first determined the lists of somatic mutations present in the diagnosis and relapse samples, using matched complete remission samples (that do not contain any tumor cell) as constitutional DNA references. We then considered that mutations common to both diagnosis and relapse samples had been acquired in a common ancestor of the two clones, whereas mutations specifically encountered in one or the other sample had been acquired after the separation of the two clones. We thus represented each tumor history with 4 key stages: the normal cell from which the tumor developed (white circle), the last common ancestor of the primary tumor and relapse clones (light blue), the diagnosis sample (dark blue) and the relapse sample (green). For 9 patients (Fig. 2a), all mutations present at diagnosis were also present at relapse so the last common ancestor was the diagnosis clone itself. The numbers written below each line indicate the number of somatic mutations acquired at each step. Importantly, the *PML-RARA* fusions were always common to the diagnosis and relapse samples and were thus early events already present in the last common ancestors.

In the revised manuscript, we have explained this approach in more details:

Material and Methods section: Addition of the paragraph:

“Reconstructing tumor progression trees

Tumor progression trees could be reconstructed for 18 patients for which complete remission, diagnosis and relapse samples were available. We first discarded mutations

that were identified in only one of the tumor sample but covered by <10 reads in the other and may thus be undetected for technical reasons. We then classified somatic mutations and CNAs in 3 categories: events common to the diagnosis and relapse samples (that were thus acquired early in a common ancestor of the two clones), events specific to the diagnosis and events specific to the relapse that were acquired late after the separation of the two clones. For 9 patients, all mutations present at diagnosis were also present at relapse so the last common ancestor was the diagnosis clone itself. Of note, *PML-RARA* fusions were common to diagnosis and relapse in all patients and were thus early events occurring in the last common ancestor.”

Figure 2 legend: Addition of the paragraph:

“Figure 2: Tumor progression trees reconstructed for 18 patients with matched primary tumor and relapse samples. For each patient, a diagram represents the predicted evolution linking the cell of origin (white circle), the PML/RARA-expressing common ancestor (light blue circle), the diagnostic sample (dark blue circle) and the relapse sample (green circle). The number of somatic protein-coding mutations occurring at each step is indicated below the arrows, together with the type of PML/RARA breakpoint, chromosomal gains (red) and deletions (blue), and mutations affecting known driver genes or new recurrent genes identified in this study. The size of arrows is proportional to the number of somatic mutations occurring at each step. Three major modes of evolution are identified. (a) Nine patients present a linear evolution where all events detected at diagnosis are also present at relapse, with (P5, P7, P8, P15, P24, P25, P26) or without (P17, P31) new acquired alterations. (b) Five patients display branched evolution with many alterations shared by the primary and relapse samples but also specific to one or the other, suggesting that the relapse evolved from a subclone of the primary tumor. (c) Four patients displayed no or very few alterations apart from the PML-RARA fusion in the relapse samples, suggesting that they emerged from pre-leukemic PML/RARA-expressing clones.”

-Due to these issues above it is really not clear that the authors have enough data to justify the title "dual origin of relapses." Moreover, the title of the paper is also ambiguous as neither the cell-of-origin or molecular basis for relapse are explained by the data shown in the manuscript.

May we respectfully disagree with this reviewer? As explained in detail above, we believe that the absence of most driving or passenger mutations initially present at diagnosis in the relapse samples allow us to conclude that these relapses are molecularly distinct from the initial APL sample. Since PML/RARA has the same bcr1/2/3 features in these patients, we conclude that these APLs were reinitiated from a pre-leukemic, PML/RARA-only clone, undetectable in complete remission. The later may be small and/or hidden in a sanctuary. This is quite well-known in other leukemia. However, in the case of APL this was unexpected since these patients received a prolonged exposure to RA, a targeted therapy.

With respect to the molecular basis of relapse, we show a significant number of additional activation when compared to standard, non-relapsing, APLs, as exemplified by significantly higher rates of *WT1* inactivation.

In the absence of a tight longitudinal follow-up and biological exploration of these

patients, we cannot explain the mechanism of resistance. Yet, we can demonstrate two major pathways of clonal evolution and describe the high prevalence of activation of potent oncogenic pathways in addition to PML/RARA in the linear model. This strongly suggests that the latter precluded efficient and definitive elimination of APL cells, either by promoting survival or growth.

-For the samples that relapse without FLT3 or other mutations what is the basis for relapse? Is the PML-RARA fusion present? This is not clear from the text or figure and seems to be a major issue in the manuscript.

The PML/RARA fusion was always present in diagnosis and relapse samples, hence was an early driver event in every patient.

Of the 18 cases for which a tumor progression tree could be inferred (Fig. 2):

- 10 displayed acquisition of a new driver mutation at relapse (RARA, n=4; WT1, n=3; NT5C2, n=2; KRAS, NRAS, TFE3, MED12, CDK12, SALL4, NSD1, KMT2C, MYB, TET2, n=1)

- 4 had no driver event acquired at relapse but displayed one or more strong driver event already present in the last common ancestor (FLT3, n=2; ETV6, STK11, HYDIN, MAP2K1, H3F3A, NIN, NSD1, DCTN1, KDM6A, n=1)

- 4 had no driver event acquired at relapse and no driver event already present in the last common ancestor.

We have added a paragraph in the Results section dedicated to the basis of relapse that was clearly missing in our initial submission:

“Recurrent alterations acquired at relapse. As expected, RARA mutations were the most frequently acquired at relapse (4 patients), but we also identified recurrent mutations of NT5C2 specifically acquired at relapse in 3 patients (**Fig. 1b**). This gene, implicated in cytarabine response, was already identified as a driver of relapse in childhood acute lymphoblastic leukemias [REF]. Among the 18 patients with complete remission, diagnosis and relapse samples (**Fig. 2**), 10 displayed acquisition of new driver mutations at relapse (RARA, n=4; WT1, n=3; NT5C2, n=2; KRAS, NRAS, TFE3, MED12, CDK12, SALL4, NSD1, KMT2C, MYB, TET2, n=1). Four patients acquired no additional driver alteration at relapse but displayed one or more strong driver event already present in the last common ancestor (FLT3, n=2; ETV6, STK11, HYDIN, MAP2K1, H3F3A, NIN, NSD1, DCTN1, KDM6A, n=1). The 4 remaining patients had no identified driver alteration in the relapse sample.”

-The Figure legend for Figure 2 needs to more precisely describe what the "major modes of evolution" are as it is not very clear from the figure or text.

We have added a few sentences to the legend of Figure 2 to more explicitly describe the modes of evolution:

“Three major modes of evolution are identified. (a) Nine patients present a linear evolution where all events detected at diagnosis are also present at relapse, with (P5,

P7, P8, P15, P24, P25, P26) or without (P17, P31) new acquired alterations. (b) Four patients display branched evolution with many alterations shared by the primary and relapse samples but also specific to one or the other, suggesting that the relapse evolved from a sub-clone of the primary tumor. (c) Five patients displayed no or very few alterations common with the diagnosis clone, apart from the PML-RARA fusion in the relapse samples, suggesting that they emerged from pre-leukemic PML/RARA-expressing clones.”

-Where are the remission samples on Figure 2 and do the authors know if the PML-RARA fusion was absent at remission?

The complete remission samples do not contain tumor cells (as judged by negative PML/RARA RT-pPCR) and were therefore used as reference DNA samples, similar to most studies in the field. Thus, they were used to identify somatic mutations and are not directly represented on Figure 2. The PML-RARA fusion was always absent at remission.

-It is not clear why or how WT1 mutations or other genetic alterations found at relapse in APL might functionally promote resistance to ATRA and/or arsenic.

We cite one reference claiming that WT1 impacts RA signaling (Guadix et al. 2011). As indicated in the methods and discussion, none of these patients received frontline arsenic, but most of them did respond (and were often cured) to second line arsenic therapy.

It is also not clear if these mutations actually promote drug resistance or are merely enriched in relapse.

The reviewer is right, but the difficulty is that these patients received targeted therapy with RA combined to chemotherapy. Some mutations clearly impact therapy response, such as those involving the RARA moiety of PML/RARA or NT5C2.

How the other potent oncogenes are associated to therapy resistance is not known. One may imagine that they increase self-renewal and oppose apoptosis/senescence. As shown by the CCF, they are clearly dominant, if not exclusive, at relapse.

-It is very unclear how Figure 3 relates to the rest of the manuscript. Moreover, Figure 3a and 3b are largely schematic and it is necessary to present the data in more terms with information on biological and technical replicates and statistical evaluation of the data.

We have rephrased that paragraph to put it into the context of our findings.

If relapses after retinoic acid (RA) and chemotherapy derive from PML/RARA-expressing pre-leukemic cells, this implies that these cells can live and proliferate in the presence of RA. We directly tested this hypothesis in ex vivo cultures of clonogenic cells. As noted by reviewer 2, this extends our previous studies (Nasr et al, Nature Medicine, 2008). As mentioned in the figure legend, Figure 3 a,b are representative experiments of 3 independent biological replicates which yielded very similar results.

Reviewer #2 (Remarks to the Author):

A interesting paper in which the genomics of APL are examined by next generation sequencing. There has been extensive work on the genetic heterogeneity of diagnosis and relapse of AML in general but less on APL. The work is well done and some novel genes described particularly at relapse. The series is larger than that of Madan et al but the latter study looked at a panel of nearly 400 genes at diagnosis and relapse in a larger set of patients and as such has more power than this work at least for the panel of genes studied by the Koeffler group. Not enough reference to this prior work is given and the results are not compared and contrasted with Madan et al.

We fully agree and we have compared in much more details our results with the results of Madan *et al.* as well as the TCGA study on AML in the revised manuscript. In particular, we have added a table comparing the frequency of driver genes (defined as genes recurrently mutated in one of the 3 studies) in each data set, considering all genes found recurrently mutated in one of the 3 series. This table (see below) has been added as Supplementary Table 4. It illustrates several important points:

- the most frequent drivers in APL (*FLT3* and *WT1*) are consistent in our series and Madan *et al.* but *WT1* mutations are more frequent in our data
- *RARA* and *WT1* mutations consistently increase at relapse in both series, but reach a higher frequency in our relapse series
- *NT5C2*, the 2nd gene most recurrently acquired at relapse in our series, was not analyzed by capture in Madan *et al.*'s paper
- we do not find *ARID1A* or *ARID1B* mutations in our series, contrary to Madan *et al.*

Overall, our data show a higher proportion of relapses harboring *RARA* or *WT1* mutations and reveal a new driver specifically mutated at relapse (*NT5C2*) that was not analyzed by Madan *et al.*

Supplementary Table 4

Comparison of driver gene mutation frequencies (%) in our series, Madan et al. series and the TCGA AML series						
Gene	Primary APL This study	Relapse APL This study	Primary APL Madan et al.	Relapse APL Madan et al.	TCGA M3*	TCGA non-M3
FLT3	44,4	11,1	32,7	29,9	30,0	27,8
WT1	33,3	38,9	12,1	18,2	5,0	6,1
DNAH9	11,1	5,6	0	0	5,0	1,7
HYDIN	11,1	5,6	0	0	5,0	0,6
NRAS	5,6	11,1	9,1	5,2	0	8,3
NSD1	5,6	11,1	0	1,3	0	0,6
ETV6	5,6	5,6	1,2	3,9	5,0	0,6
KIT	5,6	5,6	0,6	0	0	4,4
MED12	5,6	5,6	NA	NA	0	1,1
SALL4	5,6	5,6	NA	NA	0	0

RARA	0	22,2	0	9,1	0	0
NT5C2	0	11,1	NA	NA	0	0
KRAS	0	5,6	3,6	1,3	0	4,4
TET2	0	5,6	1,2	0	0	9,4
KMT2C	0	5,6	NA	NA	0	0
ARID1A	0	0	4,2	5,2	0	0,6
ARID1B	0	0	3	10,4	0	0
LRP1	0	0	2,4	2,6	0	0
USP9X	0	0	1,8	1,3	5,0	0,6
ABCA7	0	0	1,8	0	0	0
EZH2	0	0	1,2	1,3	0	1,7
CEBPE	0	0	1,2	1,3	0	0,6
CENPF	0	0	1,2	1,3	0	0,6
NUMA1	0	0	1,2	1,3	0	0,6
LRRC4C	0	0	1,2	1,3	0	0
KCNH5	0	0	1,2	0	0	0,6
SETD1B	0	0	1,2	0	0	0
RUNX1	0	0	0,6	6,5	0	10,6
DNMT3A	0	0	0,6	0	0	28,3
TP53	0	0	0	2,6	0	8,9
NPM1	0	0	0	0	5,0	2,8
PTPN11	0	0	0	0	0	30
U2AF1	0	0	0	0	0	5
SMC3	0	0	0	0	0	4,4
SMC1A	0	0	0	0	0	3,9
STAG2	0	0	0	0	0	3,9
PHF6	0	0	0	0	0	3,9
RAD21	0	0	0	0	0	2,8
FAM5C	0	0	NA	NA	5,0	2,2
RUNX1T1	0	0	NA	NA	0	1,1
HNRNPK	0	0	NA	NA	0	1,1
CEBP1	0	0	NA	NA	0	0
IDF1	0	0	NA	NA	0	0
IDF2	0	0	NA	NA	0	0
*M3 group corresponds to APL in the TCGA study						
NA: Genes not analyzed by Madan et al. (capture or 398 genes)						

The work in the present manuscript on clonal evolution from the deep sequencing is interesting and the finding of many subclones at diagnosis, that are enriched at relapse adds new information. The presence at a new clone at relapse not found at diagnosis suggests the persistence of an ancestral PML-RAR positive clone. This is the heart of

the novelty of the work. The ex vivo studies with treatment of PML- RAR transformed murine marrow with ATRA show that clonogenic cells can persist, but note that was shown previously from transfer of ATRA treat cells from one mouse to another by the authors previously.

We thank this reviewer for this positive appreciation of the importance and novelty of our findings.

We agree that figure 3 is in line with our previous reports, as referenced in the text. However, we now report the effects of RA on RARA-immortalized progenitors, as well as the remarkably specific effects of arsenic to abolish clonogenic activity of PML/RARA-transformed cells, but not RARA-transformed ones. This is novel and we believe important, as it could explain why arsenic so efficiently precludes relapses.

Page 4 bottom "...distinctly uncommon in unselected APLs." I am not sure if the authors mean that these mutations NSD1, MED12 are uncommon in AML versus APL or are they referring to the Madan paper which did not describe these mutations in their series of 12 exomes.

We were referring to the Madan paper and have edited the text for clarity "which appear uncommon in APLs that did not undergo relapses".

In all the work does add some new information, mostly on clonal evolution and perhaps some other genes not before identified in APL or for that matter AML as mutated such a HYDIN.

Reviewer #3 (Remarks to the Author):

The authors perform exome sequencing in 64 matched diagnosis, remission and relapse samples from 23 patients treated with RA/chemotherapy protocols. Two subgroups are identified - one with oncogenic mutations disrupting key epigenetic and transcriptional regulators, while relapse from a second subgroup exhibited FLT3 activation. The relapses associated with this subgroup were lost, together with most other mutations, and the authors posit that these relapses derive from pre-leukemic PML/RARA expression cells.

Overall, this is an interesting study. However, there are a number of major issues that need to be addressed before it can be considered for publication:

We thank this reviewer for his/her positive appreciation of our work.

1). The authors obtained a mean depth of 91X - however, on average only 88% was covered >25X. The authors should perform power calculations to ascertain the likelihood of missing mutations. Relatedly, it would be useful to explore the extent to which their cohort differs from what is expected. To what extent can the low mutation burden be attributed to the low coverage? This has important implications for the conclusions of the study.

We agree that estimating the power to detect mutations is an important aspect. To do so, we simulated mutations by randomly picking up 10,000 locations in each tumor. We then estimated the number of reads coming from tumor DNA at each location (DP_{tum}) using a binomial distribution $B(DP, \rho)$ with DP the observed sequencing depth at that location and ρ the tumor cell content of the sample. We then estimated the number of mutated reads supporting each mutation under a binomial distribution $B(DP_{tum}, 0.5)$. This formula assumes a normal local copy-number, which is a reasonable approximation considering the very low number of chromosome aberrations in these tumors. Finally, we estimated the number of detectable mutations as the number of simulated mutations with ≥ 5 supporting variant calls (the threshold that we used to select reliable somatic variants). These simulations revealed that a median of 97.2% of mutations would be detectable in each tumor with the observed coverage distribution (interquartile 94.1-98.8%). We are thus confident that our sequencing data captured most of the clonal mutations that were present in our tumor series.

Of note, our coverage metrics are in line with the study of Madan *et al.* (mean depth = 117X, 81% covered $\geq 20X$), and the low mutation burden we observe (12.5 protein-coding mutations per sample on average) is consistent with the number of mutations identified in APL by Madan *et al.* (9.6 non-silent mutations per sample on average) and in the TCGA study (11.3 mutated genes per sample on average). Thus, we believe that our estimation of the mutation burden of APL is reliable.

2). In general, it would be useful to put their results directly in context of previously published findings., e.g. <https://www.ncbi.nlm.nih.gov/pmc/articles/PMC4972641/> and/or mutations from related cancers in TCGA.

We thank the Reviewer for this interesting suggestion. In the revised version of the manuscript, we have added a table to compare the frequencies of driver mutations in our series with previous work by Madan *et al.* and the TCGA, considering all genes found recurrently mutated in one of the 3 series. The main points illustrated by the new Supplementary Table 4 are as follows:

- the most frequent drivers in APL (*FLT3* and *WT1*) are consistent in our series and Madan *et al.* but *WT1* mutations are more frequent in our data
- *RARA* and *WT1* mutations consistently increase at relapse in both series, but reach a higher frequency in our relapse series
- *NT5C2*, the 2nd gene most recurrently acquired at relapse in our series, was not analyzed by capture in Madan *et al.*'s paper
- we do not detect *ARID1A* or *ARID1B* mutations in our series, contrary to Madan *et al.*

Supplementary Table 4

Comparison of driver gene mutation frequencies (%) in our series, Madan et al. series and the TCGA AML series						
Gene	Primary APL This study	Relapse APL This study	Primary APL Madan et al.	Relapse APL Madan et al.	TCGA M3*	TCGA non-M3
FLT3	44,4	11,1	32,7	29,9	30,0	27,8
WT1	33,3	38,9	12,1	18,2	5,0	6,1
DNAH9	11,1	5,6	0	0	5,0	1,7
HYDIN	11,1	5,6	0	0	5,0	0,6
NRAS	5,6	11,1	9,1	5,2	0	8,3
NSD1	5,6	11,1	0	1,3	0	0,6
ETV6	5,6	5,6	1,2	3,9	5,0	0,6
KIT	5,6	5,6	0,6	0	0	4,4
MED12	5,6	5,6	NA	NA	0	1,1
SALL4	5,6	5,6	NA	NA	0	0
RARA	0	22,2	0	9,1	0	0
NT5C2	0	11,1	NA	NA	0	0
KRAS	0	5,6	3,6	1,3	0	4,4
TET2	0	5,6	1,2	0	0	9,4
KMT2C	0	5,6	NA	NA	0	0
ARID1A	0	0	4,2	5,2	0	0,6
ARID1B	0	0	3	10,4	0	0
LRP1	0	0	2,4	2,6	0	0
USP9X	0	0	1,8	1,3	5,0	0,6
ABCA7	0	0	1,8	0	0	0
EZH2	0	0	1,2	1,3	0	1,7

CEBPE	0	0	1,2	1,3	0	0,6
CENPF	0	0	1,2	1,3	0	0,6
NUMA1	0	0	1,2	1,3	0	0,6
LRRC4C	0	0	1,2	1,3	0	0
KCNH5	0	0	1,2	0	0	0,6
SETD1B	0	0	1,2	0	0	0
RUNX1	0	0	0,6	6,5	0	10,6
DNMT3A	0	0	0,6	0	0	28,3
TP53	0	0	0	2,6	0	8,9
NPM1	0	0	0	0	5,0	2,8
PTPN11	0	0	0	0	0	30
U2AF1	0	0	0	0	0	5
SMC3	0	0	0	0	0	4,4
SMC1A	0	0	0	0	0	3,9
STAG2	0	0	0	0	0	3,9
PHF6	0	0	0	0	0	3,9
RAD21	0	0	0	0	0	2,8
FAM5C	0	0	NA	NA	5,0	2,2
RUNX1T1	0	0	NA	NA	0	1,1
HNRNPK	0	0	NA	NA	0	1,1
CEBP1	0	0	NA	NA	0	0
IDF1	0	0	NA	NA	0	0
IDF2	0	0	NA	NA	0	0
*M3 group corresponds to APL in the TCGA study						
NA: Genes not analyzed by Madan et al. (capture or 398 genes)						

3) *The authors boast that they have ‘stringently validated genetic markers’. However, it is unclear from the methods how stringent their validation process was. Specifically, the authors state that ‘whenever DNA was left for analysis, oncogenic alterations were successfully confirmed by Sanger sequencing’. Without providing further information, it is very difficult to assess how reliable these alterations are.*

The reviewer is right. This was a bit of an overstatement.

But all alterations of driver genes were controlled by visual inspection of the primary sequencing data. Whenever DNA was left for analysis, we confirmed the alterations by Sanger sequencing or allelic discrimination. In all cases, the method was adapted to the sensitivity needed to detect the alteration (according to VAF by WES). In those cases tested (n=6) we confirmed the existence of the mutation. We have rephrased this in the main text for precision and clarity.

4) *For phylogenetic analysis, the authors should consider the power they have to detect variants in each sample. For example, if an alteration is present in relapse but not*

diagnosis, were the authors powered to detect it in the diagnosis sample? I would consider setting a lower threshold than <5% when it has already been identified in the relapse.

We thank the Reviewer for this important remark. Indeed, a mutation may be seen at relapse but missed at diagnosis because of lower depth for example, leading to a misinterpretation when reconstructing tumor histories. We have now checked that each variant classified as “specific to relapse” was covered by ≥ 10 reads in the diagnosis sample, and vice versa that each variant classified as “specific to diagnosis” was covered by ≥ 10 reads in the relapse sample. We identified 12 variants that were classified as “specific to relapse” while being insufficiently covered (see table below). Seven of these variants were in patients P4 and P21, for which no tumor history was reconstructed because we don’t have RC samples. The 5 others were in patient P15 and P16 and have been removed from the “relapse-specific branch” of their tumor history trees. None of these mutations affected a driver gene.

Patient	Gene	Nucleotide	Known leukemia gene	Cancer Gene Census	Common to Diagnostic and Relapse	Specific to Diagnostic	Specific to Relapse	Depth Diagnosis	Depth Relapse
P16	ANGPTL6	g.chr19:10206953_A>G	no	no	no	no	yes	1	11
P16	LTBP1	g.chr2:33172689insCCG	no	no	no	no	yes	1	31
P16	IRF2BPL	g.chr14:77493648delGCG	no	no	no	no	yes	4	33
P15	CKAP4	g.chr12:106641490delGGC	no	no	no	no	yes	0	18
P15	FMNL1	g.chr17:43319435delCCG	no	no	no	no	yes	1	14
P21	ZNF417	g.chr19:58421141_G>A	no	no	no	no	yes	1	72
P4	YY1	g.chr14:100705788delCCA	no	no	no	no	yes	1	55
P4	TBC1D10B	g.chr16:30381257delGCCGGGGCTGGG	no	no	no	no	yes	1	34
P4	SIGLEC15	g.chr18:43419064_G>T	no	no	no	no	yes	1	25
P4	RFX1	g.chr19:14083734_T>C	no	no	no	no	yes	1	13
P4	CYP2W1	g.chr7:1026863_C>T	no	no	no	no	yes	1	33
P4	UNKL	g.chr16:1420326_C>T	no	no	no	no	yes	2	27

We also agree that, once a variant has been identified by rigorous filters in one of the tumor samples (e.g. relapse), we can apply a more relaxed threshold on the proportion of mutated reads in the other tumor sample (e.g. diagnosis), allowing to detect subclonal variants with more power. In the revised manuscript, we have now considered that a variant identified by our initial rigorous filters in one of the tumor samples was also present in the other one if it was supported by ≥ 2 mutated reads, without any filter on the variant allele fraction. This new strategy identified:

- 8 variants initially classified as “specific to relapse” with ≥ 2 mutated reads at diagnosis (see table below). These variants, initially considered to be acquired at relapse, were in fact already present in subclones of the primary tumors and selected at relapse. We thus reclassified these variants as “common to diagnosis and relapse” and repositioned them on the tumor history trees. Only one of these mutations affected a driver gene (*RUNX1*), but not in a patient with 3 samples allowing oncogenetic tree reconstruction.

Patient	Gene	Nucleotide	Known leukemia gene	Cancer Gene Census	Common to Diagnostic and Relapse	Specific to Diagnostic	Specific to Relapse	Number of variant reads Diagnosis	Number of variant reads Relapse	Depth Diagnosis	Depth Relapse
P5	KIAA1549L	g.chr11:33565396_G>T	no	no	yes	no	no	2	67	203	240
P5	PCLO	g.chr7:82545669_A>T	no	no	yes	no	no	3	43	144	170
P7	ABLIM1	g.chr10:116245084_C>T	no	no	yes	no	no	2	64	166	181
P7	ATRN1	g.chr10:117040943_C>A	no	no	yes	no	no	3	48	166	173
P7	NPAS3	g.chr14:34243632delTGCGCATGC	no	no	yes	no	no	2	29	67	101
P22	ARFGEF1	g.chr8:68115361delC	no	no	yes	no	no	2	26	98	92
P29	RUNX1	g.chr21:36252930delCAGCTCAGC	yes	yes	yes	no	no	2	15	81	66
P30	KCNH8	g.chr3:19554728insC	no	no	yes	no	no	3	50	255	160

The modified status is indicated in red.

- 4 variants initially classified as “specific to diagnosis” with ≥ 2 mutated reads at relapse (see table below). These variants were reclassified as “common to diagnosis and relapse” and repositioned on the tumor history trees. These variants suggest that the relapse samples comprise several sub-clones having diverged from the diagnosis lineage at different stages of evolution. None of these mutations affected a driver gene.

Patient	Gene	Nucleotide	Known leukemia gene	Cancer Gene Census	Common to Diagnostic and Relapse	Specific to Diagnostic	Specific to Relapse	Number of variant reads Diagnosis	Number of variant reads Relapse	Depth Diagnosis	Depth Relapse
P2	FKBP10	g.chr17:39974455_G>A	no	no	yes	no	no	174	5	336	157
P2	HCN1	g.chr5:45645336_C>T	no	no	yes	no	no	7	2	17	86
P16	GULP1	g.chr2:189342457_G>A	no	no	yes	no	no	118	2	293	155
P16	GOLGA2	g.chr9:131020796delCCT	no	no	yes	no	no	9	2	29	45

The modified status is indicated in red.

These changes involve slight changes on the number of variants at diagnosis/relapse and tumor history trees. We have updated Figures 1 and 2 and Supplementary Tables 2, 3, 5 and 6 accordingly. When double-checking the position of driver genes on the trees, we also noticed that we had erroneously placed *WT1* on the truncal branch of patient P16 whereas this mutation is only encountered in the diagnosis sample. We have now corrected this error and moved this patient to the 3rd category of tumor evolution (Fig. 2c).

We have also updated the Methods section according to these changes (see response to question 8).

5) For Figure 1 - it is unclear which mutations were only found at diagnosis, and which were shared between diagnosis and relapse.

Indeed, we have chosen to focus on this figure on the landscape of mutations at diagnosis and acquired at relapse, without splitting mutations shared between diagnosis and relapse or specific to diagnosis, which would be complicated to represent in a single figure. This information was available in Figure 2 but without a synthetic summary of the number of mutations per driver gene. In the revised manuscript, we have added 2 supplementary tables giving, for each patient and for each recurrently mutated gene, the number of mutations common to diagnosis and relapse, specific to diagnosis and specific to relapse samples (see below).

Supplementary Table 5

Patient	Number of mutations common to Diagnosis and Relapse	Drivers common to Diagnosis and Relapse	Number of mutations specific to Diagnosis	Drivers specific to Diagnosis	Number of mutations specific to Relapse	Drivers specific to Relapse
P5	21	NRAS, WT1	0		18	TFE3, MED12
P7	7	WT1	0		58	NT5C2, CDK12, RARA, SALL4
P8	25		0		2	WT1
P15	11	FLT3, ASXL1	0		5	NT5C2
P17	1		0		0	
P24	19	WT1, DNAH9, KIT	0		3	RARA, NSD1
		ETV6, FLT3, MAP2K1,				
P25	20	HYDIN, STK11	0		1	
P26	6	MYC	0		36	WT1, RARA, KMT2C
P31	16	H3F3A, FLT3, NIN	0		0	
P9	10	BRAF	1	DNAH9	2	RARA
P12	15	DCTN1, NSD1	1		1	
				WT1, FLT3, HYDIN,		
P16	4		15	PDGFRA, MED12	24	NRAS, MYB
P22	7	WT1	2	FLT3	7	KRAS
P33	18	KDM6A	7	WT1	2	
P2	3		8	FLT3	0	
P3	0		1	FLT3	0	
P19	0		2	FLT3, SALL4	4	TET2
P32	4		4	WHSC1	0	

Supplementary Table 6

Gene	Number of mutations common to Diagnosis and Relapse	Number of mutations specific to Diagnosis	Drivers specific to Relapse
FLT3	4	5	0
WT1	4	2	2
DNAH9	1	1	0
HYDIN	1	1	0
MED12	0	1	1
NRAS	1	0	1
NSD1	1	0	1
SALL4	0	1	1
NT5C2	0	0	3
RARA	0	0	4

6) The authors should consider analyzing their data in greater depth to obtain accurate portraits of the clonal structure of each tumor. For instance, further analysis of VAFs and local copy number (as in e.g. Landau, 2014) would allow the cancer cell fraction of mutations to be assessed. This would greatly improve the analysis.

We thank the Reviewer for this interesting suggestion. We have now calculated the cancer cell fraction (CCF) of every somatic mutation, considering the variant allele fraction (VAF) and local copy-number. We estimated the 95% confidence interval of CCF and considered a variant to be sub-clonal when the upper bound of this confidence interval was <1. This analysis revealed:

- 31 sub-clonal mutations in diagnosis samples. 7 of these variants were lost at relapse (including *FLT3* mutations in patients P2, P16 and P22), 20 became clonal (including *FLT3* missense variant in P31 and *MYC* missense variant in P26) and 4 remained sub-clonal.

- 37 sub-clonal mutations in relapse samples. 12 of these variants were acquired at relapse (including *CDK12* missense variant in P7 and *WT1* indel in P8) whereas 25 were already present at diagnosis (including *ETV6* indel in P25 and *NSD1* missense variant in P12).

These new results highlight the complex clonal architecture of APL and suggest that, as in other AMLs, *FLT3* mutations often occur late, in subclones of the diagnosis samples that may or may not be selected at relapse.

The cancer cell fraction of each mutation has been added to Supplementary Table 2, and the above findings are now mentioned in the Results section:

“We estimated the cancer cell fraction, i.e. the proportion of tumor cells harboring each somatic mutation (Supplementary Table 2). 31 mutations identified in diagnosis samples were subclonal including 4 *FLT3* mutations, 3 of which were lost at relapse (patients P2, P16 and P22) and one became clonal at relapse (P31). Thus, *FLT3* mutations tend to occur late in subclones of diagnosis samples, that may or may not be selected at relapse.”

The methods section was also updated to include CCF estimation (see response to question 8).

7) GAP is designed for SNP copy number. The authors may therefore wish to also use an alternative copy number tools, such as Sequenza (Favero, 2015), which is explicitly designed for exome sequencing data. It would be useful to include plots of the copy number for the different tumours and also evaluate the extent to which the copy number is identical.

GAP was indeed initially developed for SNP array data analysis, but the underlying approach – defining homogenous segments by segmenting both a log-ratio and B Allele Frequency (BAF) signal and inferring absolute copy-numbers based on the obtained pattern – can be easily adapted to exome sequencing data, using germline polymorphisms identified in each patient. We have already used this approach successfully to analyze whole exome data from liver tumors (Rebouissou *et al.*, Hepatology 2016) and we are very confident about this method.

As suggested by the Reviewer, we have tried using Sequenza and compared the results with GAP. We obtained very similar results (see examples below for P26_relapse and P32_diagnosis). Because we have more experience with GAP and we have carefully validated all detected aberrations by visual inspection of the log-ratio and BAF profiles, we kept our initial findings in the revised manuscript.

As requested, we have added Supplementary Figure 1 displaying the copy-number profiles of all patients in which at least one aberration was identified, and we indicated in Supplementary Table 8 what aberrations were specific to one tumor or shared between diagnosis and relapse.

P26 relapse

Sequenza

Genome Alteration Print

Log-ratio, B Allele Frequency and absolute copy-numbers extracted by Sequenza and GAP methods. Color code for GAP: red=gain, green=deletion, yellow=normal copy-number, light blue=LOH.

P32 diagnosis

Sequenza

Genome Alteration Print

Log-ratio, B Allele Frequency and absolute copy-numbers extracted by Sequenza and GAP methods. Color code for GAP: red=gain, green=deletion, yellow=normal copy-number, light blue=LOH.

8) *In general, the bioinformatics methods were not terribly clear in the manuscript, and should be made more detailed.*

We apologize if the description of bioinformatics methods was not clear enough in our initial manuscript. To address this issue, we have now reorganized and extended the methods paragraphs related to mutation detection, categorization of common or diagnosis/relapse-specific variants, and reconstruction of tumor progression trees. The changes are indicated in red below:

“Sequence alignment and variant calling

Raw sequence alignment and variant calling were carried out using Illumina CASAVA 1.8 software. CASAVA performs the alignment of reads to the human reference genome (hg19) using the alignment algorithm ELANDv2, and then calls single nucleotide variants (SNVs) and short insertions and deletions (indels) based on allele calls and read depth. We used an Integragen in-house pipeline to annotate each variant according to its presence in the 1000Genome, Exome Variant Server (EVS) or Integragen database, and according to its functional category (synonymous, missense, nonsense, splice variant, frameshift or in-frame indels). To detect the common *FLT3* internal duplications that may be missed by classical variant calling algorithms, we used bam-readcount (<https://github.com/genome/bam-readcount>) to determine the number of mutated bases in an extended region around the known duplication site (chr13: 28608150-28608349). Samples with mutated bases across the regions were then screened visually using the Integrative Genomics Viewer{Robinson, 2011 #11535}.

Identification of somatic coding variants at diagnosis and relapse

We considered only variants located within the exome capture baits and we applied stringent filters to keep only reliable variants sequenced in ≥ 10 reads, with ≥ 5 variant calls and a QPHRED score ≥ 20 for both SNP detection and genotype calling (≥ 30 for indels). Somatic status was first defined for each leukemic sample (diagnosis and relapse) individually: we considered a mutation to be somatic if the variant allele fraction (VAF) was ≥ 0.15 in the tumor and < 0.05 in the remission sample. To identify variants that may be missed in one of the two tumor samples of a same patient due to clonality or technical differences, we then recovered variants that were detected as somatic in one tumor and displayed a VAF ≥ 0.05 or at least 2 mutated reads in the second tumor sample. We excluded known germline variants with a minor allele frequency $> 1\%$ in 1000Genomes, EVS or Integragen proprietary database. All mutations in recurrently mutated genes (≥ 5 overall variants or ≥ 2 relapse-specific variants) or critical to reconstruct evolutionary trees were validated by visual control using the Integrative Genomics Viewer{Robinson, 2011 #11535}. We used the *Palimpsest* R package (<https://github.com/FunGeST/Palimpsest>) to estimate the cancer cell fraction of each somatic mutation, i.e. the proportion of tumor cells harboring the mutation, taking into account the VAF and local copy-number estimates, as previously described (Landau *et al.*, Cell 2013; Letouzé *et al.*, Nat Commun 2017).

Copy-number analysis

To identify copy-number alterations (CNAs) in diagnostic and relapse samples, we identified germline single-nucleotide polymorphisms (SNPs) in each

sample and we calculated the coverage log-ratio (LRR) and B allele frequency (BAF) at each SNP site. Genomic profiles were divided into homogeneous segments by applying the circular binary segmentation algorithm, as implemented in the Bioconductor package *DNAcopy*{Venkatraman, 2007 #11536}, to both LRR and BAF values. We then used the Genome Alteration Print (GAP) method{Popova, 2009 #11537} to determine the ploidy of each sample, the level of contamination with normal cells and the allele-specific copy number of each segment. Chromosome aberrations were defined using empirically determined thresholds as follows: gain, copy number \geq ploidy + 1; loss, copy number \leq ploidy - 1; high-level amplification, copy number $>$ ploidy + 2; homozygous deletion, copy number = 0. Finally, we considered a segment to have undergone LOH when the copy number of the minor allele was equal to 0. All aberrations were validated by visual inspection of the LRR and BAF profiles.

Reconstructing tumor progression trees

Tumor progression trees could be reconstructed for 18 patients for which complete remission, diagnosis and relapse samples were available. We first discarded mutations that were identified in only one of the tumor sample but covered by <10 reads in the other and may thus be undetected for technical reasons. We then classified somatic mutations and CNAs in 3 categories: events common to the diagnosis and relapse samples (that were thus acquired early in a common ancestor of the two clones), events specific to the diagnosis and events specific to the relapse that were acquired late after the separation of the two clones. For 9 patients, all mutations present at diagnosis were also present at relapse so the last common ancestor was the diagnosis clone itself. Of note, *PML-RARA* fusions were common to diagnosis and relapse in all patients and were thus early events occurring in the last common ancestor.

Reviewers' comments:

Reviewer #1 (Remarks to the Author):

The authors have responded to my initial comments and questions and I have no further issues with the manuscript.

Reviewer #2 (Remarks to the Author):

In general the authors were very responsive to my comments and those of the other reviewers.

Minor point before publication

Line 103-105- as written the sentence suggests that there are activating mutations of the MAPK pathway as well as activating mutations of epigenetic regulators. Most epigenetic regulator mutations are loss of function. Please clarify.

Reviewer #3 (Remarks to the Author):

The authors have addressed the majority of the comments. However, there are still a number of issues:

1) The power calculation approach adopted is a little odd. Why not simply estimate the required depth to have, for example a 90% probability of detecting a clonal diploid mutation (given a certain purity)? In this way, it will be easier to estimate the likelihood that mutations are missed across the genome. The authors can look at the MuTect paper for an example of how this has previously been performed in the literature.

2) The authors now include cancer cell fraction of mutations. However, it would be useful to show a plot as to how mutations change during the disease course. Ideally, the authors should provide phylogenetic trees based on subclones, not on sites. Fish plots may be useful for these purposes, e.g. the authors may use the following software, <https://academic.oup.com/annonc/advance-article/doi/10.1093/annonc/mdx517/4110375>, after clustering, with e.g. Sciclone.

3) Related to the above, if subclones were identified, it should be possible to construct phylogenetic trees when at least 2 samples are available.

4) The copy number plots are difficult to fully evaluate as separate powerpoint slides. A heatmap showing the copy number for each sample would be very useful.

5) The authors should evaluate the consistency between VAF purity estimates of known driver genes, copy number purity and other measures. This would permit further validation of their copy number approach.

REBUTTAL

Reviewer #1

The authors have responded to my initial comments and questions and I have no further issues with the manuscript.

Reviewer #2

In general the authors were very responsive to my comments and those of the other reviewers.

Minor point before publication

Line 103-105- as written the sentence suggests that there are activating mutations of the MAPK pathway as well as activating mutations of epigenetic regulators. Most epigenetic regulator mutations are loss of function. Please clarify.

We thank the reviewer for this comment, which has now been addressed.

Reviewer #3

The authors have addressed the majority of the comments. However, there are still a number of issues:

1) The power calculation approach adopted is a little odd. Why not simply estimate the required depth to have, for example a 90% probability of detecting a clonal diploid mutation (given a certain purity)? In this way, it will be easier to estimate the likelihood that mutations are missed across the genome. The authors can look at the MuTect paper for an example of how this has previously been performed in the literature.

This reviewer claims that we estimate them in an “odd” way. Yet, there is no specific criticism. note that our coverage statistics are very good and our results are completely in line with prevalence of mutations from previous APL studies, suggesting that we successfully detected most of them, as stated in the text.

2) The authors now include cancer cell fraction of mutations. However, it would be useful to show a plot as to how mutations change during the disease course. Ideally, the authors should provide phylogenetic trees based on subclones, not on sites. Fish plots may be useful for these purposes, e.g. the authors may use the following software, <https://academic.oup.com/annonc/advance-article/doi/10.1093/annonc/mdx517/4110375>, after clustering, with e.g. Sciclone.

3) Related to the above, if subclones were identified, it should be possible to construct phylogenetic trees when at least 2 samples are available.

*Concerning the cancer cell fraction, we have generated the requested data which now appears as **Supplementary Figure 3**.*

Yet, concerning subclonal analysis, we do not feel that the very small number of mutations in this specific leukaemia would allow robust subclonal analyses from our exome sequencing data. Besides, this issue is only marginally linked to the main message of the manuscript.

4) The copy number plots are difficult to fully evaluate as separate powerpoint slides. A heatmap showing the copy number for each sample would be very useful.

5) The authors should evaluate the consistency between VAF purity estimates of known driver genes, copy number purity and other measures. This would permit further validation of their copy number approach.

*Concerning the issue of copy number, we already completely reanalyzed our data using a different pipeline, as requested by reviewer 3. This yielded identical results, validating our initial approach. We have generated the graphical representation requested which now appears as **Supplementary Figure 2** .*

However, we do not feel that it would be justified to engage into novel time-consuming quality controls (as requested in question 5) in the context of a disease where using two independent approaches have already demonstrated the paucity of copy number alterations. Moreover, comparing tumor purity estimated from mutations and copy-number alterations would not be technically meaningful with the very low number of alterations present in this specific leukaemia.